# Using Respiratory Gas Analyzers to Determine Resting Metabolic Rate in Adults: A Systematic Review of Validity Studies

**DOI:** 10.3390/sports13070198

**Published:** 2025-06-22

**Authors:** César Ulises Olivas-León, Francisco Javier Olivas-Aguirre, Isaac Armando Chávez-Guevara, Horacio Eusebio Almanza-Reyes, Leslie Patrón-Romero, Genaro Rodríguez-Uribe, Francisco José Amaro-Gahete, Marco Antonio Hernández-Lepe

**Affiliations:** 1Conahcyt National Laboratory of Body Composition and Energetic Metabolism (LaNCoCoME), Tijuana 22390, Mexico; cesar.olivas@uabc.edu.mx (C.U.O.-L.); javier.olivas@uacj.mx (F.J.O.-A.); isaac.chavez.guevara@uabc.edu.mx (I.A.C.-G.); almanzareyes@uabc.edu.mx (H.E.A.-R.); leslie.patron@uabc.edu.mx (L.P.-R.); genaro.rodriguez@uabc.edu.mx (G.R.-U.); 2Faculty of Medicine and Psychology, Autonomous University of Baja California, Tijuana 22390, Mexico; 3Department of Health Sciences, Biomedical Sciences Institute, Autonomous University of Ciudad Juarez, Ciudad Juarez 32310, Mexico; 4Faculty of Sports Ensenada, Autonomous University of Baja California, Ensenada 22800, Mexico; 5Department of Medical Physiology, Faculty of Medicine, Sport and Health University Research Institute (iMUDS), University of Granada, 18001 Granada, Spain; amarof@ugr.es; 6CIBER Physiopathology of Obesity and Nutrition, Institute of Health Carlos III, 28029 Granada, Spain; 7Biosanitary Research Institute, Ibs.Granada, 18012 Granada, Spain

**Keywords:** energetic expenditure, indirect calorimetry, validation study, reproducibility of results

## Abstract

Background: Correct assessment of resting metabolic rate (RMR) is fundamental for estimating total energy expenditure in both clinical nutrition and sports sciences research. Various methods have been proposed for RMR determination, including predictive equations, isotopic dilution techniques, and indirect calorimetry. Over the past two decades, portable gas analyzers have emerged as promising alternatives, offering more accessible and cost-effective solutions for metabolic assessment. However, evidence regarding their validity remains inconsistent, particularly across diverse populations and varying metabolic assessment protocols. Methods: This systematic review was conducted in May 2025 using the PubMed, Web of Science, and EBSCO databases, following the PRISMA-DTA guidelines, and included observational studies with the objective of examining the available evidence regarding the validity of portable gas analyzers to determine RMR in humans. The methodological quality of each study was assessed using the NIH Quality Assessment Tool for Observational Cohort and Cross-Sectional Studies. Results: From an initial pool of 230 studies, 16 met the eligibility criteria. The findings revealed notable variability in measurement validity among devices, mainly influenced by device model, population characteristics, and methodological factors. While portable analyzers such as FitMate and Q-NRG exhibited high validity, MedGem exhibited systematic biases, particularly in individuals with higher adiposity, leading to RMR overestimations. Conclusions: The main results demonstrated the critical need for rigorous validation of portable gas analyzers before their implementation in clinical and research settings to ensure their applicability across diverse populations and metabolic assessments.

## 1. Introduction

The proper assessment of resting metabolic rate (RMR) is essential for estimating total energy expenditure in clinical nutrition and sports sciences research [1]. Indeed, an accurate measure is crucial for (i) developing personalized dietary interventions, (ii) optimizing metabolic health, and (iii) managing conditions such as obesity, diabetes, and malnutrition [2]. In this context, bias in this estimation can lead to inadequate caloric prescriptions, potentially affecting nutritional status, weight management, and clinical outcomes [3]. Consequently, the establishment of robust and standardized methodologies for RMR measurement remains a critical objective in both research and clinical practice [4].

One of the earliest methods developed to determine whole-body metabolic rate was the Douglas bag, which consists of evaluating expired gases collected from an exhaled breath over a specific period in a twill bag lined with vulcanized rubber. It is considered the gold standard for measuring respiratory gas exchange but has some limitations such as a limited sample due to the test time for each subject, air leaks or condensation in the bags, and the lack of a standard reference for validation [5].

Several approaches are commonly used to estimate RMR, including predictive equations, but this method does not provide direct measurements of RMR. Instead, indirect calorimetry is often used to determine RMR, which is considered a reference standard for measuring metabolic gas exchange, as it provides direct measurements of oxygen consumption (VO_2_) and carbon dioxide production (VCO_2_) under controlled resting conditions [6,7]. Traditionally, metabolic carts have been used for this purpose using different sensor technologies, calibration procedures, or data-processing algorithms. However, variations in these methodologies make it difficult to standardize indirect calorimetry protocols across different clinical and research settings [8]. Furthermore, metabolic carts require controlled laboratory conditions, limiting their feasibility for widespread use in non-laboratory settings.

To address these constraints, portable gas analyzers have been developed over the past two decades, offering a practical and accessible approach for assessing RMR [9]. However, the validity of these devices has been mostly investigated among athletic populations, where metabolic efficiency is a key performance factor [10,11]. Indeed, Van Hooren et al. [12] recently reported that VO2masterPro underestimated VO_2_ by an average of ~12%, while PNOĒ overestimated VO_2_ by an average of ~8.3%, being less accurate than stationary metabolic carts for assessing energy expenditure.

Currently, there is scarce information about the validity of portable gas analyzers in the general population, including in healthy individuals and patients with metabolic disorders. If rigorous validation and standardization are not thoroughly conducted, the use of portable gas analyzers may lead to the misdiagnosis of metabolic dysfunctions, the development of ineffective treatment plans, and the generation of inaccurate research findings. Therefore, this work aimed to systematically examine the available evidence regarding the validity of portable gas analyzers in humans across diverse populations and clinical settings.

## 2. Materials and Methods

This systematic review was strictly conducted in full accordance with the elements outlined in the Preferred Reporting Items for a Systematic Review and Meta-analysis of Diagnostic Test Accuracy Studies: The PRISMA-DTA Checklist (Appendix A) [13]. Furthermore, the protocol and its methodological considerations were registered in the International Prospective Register of Systematic Reviews (PROSPERO) public database (ID: CRD420250652077).

### 2.1. Eligibility Criteria

The included articles in this work were full-text observational studies that assessed RMR in humans. The specific characteristics of the studies were determined using the following PICO strategy: (i) Participants: Healthy untrained adults (>18 years). (ii) Intervention: Measurement of RMR using different portable gas analyzers. (iii) Comparison: Similarities and differences between multiple devices or across different models/equipment. Outcomes: Validity of portable gas analyzers for assessing RMR.

### 2.2. Exclusion Criteria

After the removal of duplicates, manuscripts with any of the following characteristics were excluded: (i) studies involving patients with pre-existing cardiorespiratory or metabolic disorders, (ii) studies using predictive equations to estimate RMR, (iii) systematic review articles, abstracts, letters to the editor, and conference proceedings, and (iv) studies that lacked clearly defined protocols in their design for RMR measurement.

### 2.3. Information Sources

The literature search was conducted using three major scientific databases: PubMed, Web of Science, and EBSCO. The following search string with Boolean operators was applied: Adults AND (“resting metabolic rate” OR “resting energy expenditure” OR “basal energy expenditure” OR “RMR” OR “basal metabolic rate” OR “BMR”) AND (“portable metabolic analyzer” OR “metabolic cart” OR “respiratory gas analyzer” OR “indirect calorimetry” OR “metabolic analyzer”) AND (“comparison” OR “validation” OR “consistency” OR “agreement” OR “reliability” OR “accuracy”) NOT equations. The search was conducted on May 2025 and the retrieved results were further restricted to studies published between 2000 and 2025.

### 2.4. Data Collection and Evaluation of Methodological Quality/Risk of Bias

Data extraction and verification were conducted independently by two investigators (C.U.O.-L. and M.A.H.-L.). The primary outcomes included bibliographic information, participant age, evaluated devices, measured RMR values, and study conclusions. The methodological quality of each study was assessed using the NIH Quality Assessment Tool for Observational Cohort and Cross-Sectional Studies [14]. The risk of bias was de-fined through providing a value of yes/no to 14 questions, resulting in regular (≤10/14 yes) or low (≥11/14 yes) risk of bias. Methodological quality evaluation was conducted independently by two reviewers (C.U.O.-L. and M.A.H.-L.), and any discrepancies were resolved through consensus.

## 3. Results

A total of 364 articles were initially retrieved from PubMed (*n* = 143), Web of Science (*n* = 106), and EBSCO (*n* = 115). After removing 158 duplicates, 206 unique studies proceeded to further screening by title, abstract, and keywords. A total of 173 studies were excluded based on predefined criteria: inappropriate study design (systematic reviews, *n* = 2), irrelevant outcomes (*n* = 162), animal studies (*n* = 1), and population characteristics misaligned with inclusion criteria (*n* = 8). Following this phase, 33 studies were entirely read, excluding 15 additional papers due to either the absence of a standard reference method or the lack of direct RMR assessment. Lastly, 18 studies met inclusion criteria and were included in the final analysis (Figure 1).

The main characteristics of the studies are described in Table 1, where it is shown that the validity of metabolic measurement devices differs across studies, with some demonstrating strong agreement with reference methods and others presenting systematic biases influenced by population characteristics.

Finally, the methodological quality/risk of bias of the selected studies is described in Table 2. Most of them showed a “Regular” risk of bias (n = 11), while only 7 were classified as having a “low” risk of bias. Overall, scores related to population sampling, recruitment of a representative sample, and repeated assessment of the exposure factor (evaluation using gas analyzers) were low in most studies.

Devices such as FitMate, Q-NRG, and the Pocket-Sized Metabolic Analyzer have shown high accuracy in assessing VO_2_ and VCO_2_ when compared to the Douglas Bag method, a widely recognized gold standard for metabolic gas analysis. Notably, the Q-NRG and FitMate demonstrated high accuracy when comparing their VO_2_ and VCO_2_ measurements to gold-standard methodologies, such as the Douglas Bag method. However, some devices exhibit systematic biases, particularly in populations with varying levels of adiposity and body composition, which can complicate their applicability in a broader clinical context. For instance, MedGem, while widely used due its portability and ease of use, tends to overestimate RMR in individuals with obesity, potentially due its simplified measurement algorithm failing to capture metabolic variability within heterogeneous populations [15,30]. In contrast, FitMate has shown high reproducibility and agreement with gold-standard methods. Nieman et al. [22], reported no statistically significant differences between FitMate and the Douglas Bag method for VO_2_ measurements (242 ± 49 mL/min vs. 240 ± 49 mL/min, *p* = 0.066) and RMR (1662 ± 340 kcal/day vs. 1668 ± 344 kcal/day, *p* = 0.579), reinforcing its reliability.

While the Douglas Bag method remains the gold standard for metabolic analysis, its use in validation studies is limited. Only 16.66% of the reviewed studies [22,32] employed the Douglas Bag method, with 83.33% relying on metabolic carts such as Delta-Trac and Quark RMR.

Additionally, variations in measurement protocols, including subject positioning (seated vs. supine) and evaluations duration (ranging from 5 to 30 min), introduce further discrepancies across studies. Differences in pre-test resting time and measurement duration, often dictated by manufacturer recommendations, may contribute to inconsistencies in reported RMR values. The Q-NRG device, compared to Delta-Trac, Quark RMR, and V-max, reported a shorter measurement time and showed consistency in RMR measurements across sessions, reinforcing its potential for repeated assessments. However, its performance may be influenced by factors such as the use of the face mask vs. the canopy hood [25].

## 4. Discussion

The present systematic review was designed to examine the existing evidence regarding the reliability, validity, and accuracy of portable gas analyzers in humans. The results show only 18 studies that present scientific evidence of good methodological quality across diverse populations and clinical settings using a reference standard, resulting in significant differences in the reliability, validity, and accuracy of portable gas analyzers in measuring different metabolic parameters across various devices and study populations.

Zhao et al. [32] reported a strong correlation between the Pocket-Sized Metabolic Analyzer and the Douglas Bag method, although an approximate 10% variation in results was observed. While this discrepancy is relatively small, it highlights the potential influence of minor differences in device calibration and methodology.

Studies such as Frankenfield & Coleman [19] and Compher et al. [16] indicated that MedGem systematically overestimated RMR in individuals with obesity when compared to Delta-Trac, a widely used metabolic cart. Interestingly, this overestimation was not observed in non-obese individuals, suggesting that body composition significantly influences MedGem’s accuracy.

Other devices, such as FitMate GS and Q-NRG, have demonstrated variability in accuracy when compared to whole body calorimetry and other metabolic carts. In particular, Purcell et al. [24] reported that FitMate GS underestimated RMR, although it exhibited high reproducibility across sessions, which supports its potential use in settings where consistency is prioritized over absolute accuracy. Additionally, Dupertuis et al. [26] found that Q-NRG’s accuracy depends on the use of a canopy hood or a face mask, with the hood mode providing more reliable VO_2_ and VCO_2_ measurements than the mask mode, which overestimated RMR in men. This variability emphasizes the importance of selecting the appropriate application mode when using Q-NRG in clinical or research settings.

Of the included studies only three used the Douglas Bag (gold standard) as the validation method for metabolic analysis, with a total of 15 of the included studies relying on metabolic carts as the validation method. While these devices are commonly used, their inherent variability in accuracy raises concerns regarding cross-device standardization and comparability [33,34]. The limited use of the Douglas Bag method highlights the need for further validation studies using gold-standard techniques to improve measurements consistency.

There were only two studies found with populations with risk factors of metabolic diseases; Purcell et al. [24] reported differences in evaluated RMR in normal/overweight subjects compared to obesity, and Hlynsky et al. [17] evaluated women with anorexia, reporting statistical differences of gas analyzers at comparing this population with a control group. It is important to mention that it has been reported that metabolic diseases can affect RMR by either increasing or decreasing it, making evaluation difficult [35], and there exists a need to implement studies focused on describing the best equipment or method to avoid bias when evaluating energetic metabolism in subjects with metabolic diseases.

Currently, there are various devices on the market claiming to be of high quality for assessing metabolic gas exchange; however, there is little evidence of their validity and reproducibility, and the existing evidence is focused on specific populations such as high-performance athletes and hospitalized patients. An example of this is the BIOPAC GASSYS3, which presents significant differences in its measurements for RMR in specific populations, requiring the development of equations to correct the data, which calls into question its validity as an assessment device [36].

Devices such as Delta-Trac and VO2000 have proven their validity compared to the standard methods; however, their manufacture is discontinued, making them impossible to acquire. Likewise, there are devices that have implemented technological improvements compared to previous versions, such as Cortex MetaMax 3B or Cosmed K5. Although these updates are nearly a decade old, these devices remain reliable, but low demand and high costs could hamper the development of new devices of this type [37]. In addition, several portable systems have been developed that are designed to provide basic VO_2_ measurement and RMR estimation. However, these devices have limitations, as they only provide O_2_ analysis and require assumptions about RMR. These portable devices are unlikely to be accepted in high-quality research where direct measures of these variables are required [38].

The principal limitations of the discussed documents include the low sample size (≤60 participants in 77.8% of the included studies), the large variation in the ages of the participants, and the fact that 44.44% used the gas analyzer Delta-Trac as the validation method; this device has been discontinued so is not available to be obtained in the market. However, additional variables applicable in both clinical and sports settings could be explored for metabolic evaluations purposes, such as VO_2_max or maximal fat oxidation, not only variables related to RMR [39]. Furthermore, because the included studies used distinct metrics to represent the validity of the evaluated devices (i.e., mean differences, LoA, ICC, etc.), this made it difficult to compare the results of the studies. Moreover, it was not possible to pool all the data in a robust meta-analysis due to the lack of available data for the Pocket-Sized Metabolic Analyzer, IIM-IC-100 VO2000, and Ecovx Beacon. Finally, the analyzed findings suggest that the characteristics of the studied populations, the technology of each device, the type of measurement, the sampling method, the test range, and the calibration method can influence the accuracy of device measurements, highlighting the importance of conducting studies following the statistical guidelines of Hopkins et al. [40] in diverse populations to standardize the assessment of the validity of portable gas analyzers.

## 5. Conclusions

This review underscores the variability in reliability and accuracy among respiratory gas analyzers, emphasizing the need for rigorous validation studies. While some devices align closely with gold-standard methods, others demonstrate systematic biases influenced by population characteristics and measurement protocols. A key finding is that only 16.66% of the reviewed studies utilized the Douglas Bag method, with the majority relying on metabolic carts. While these carts are widely used, their variability raises concerns about cross-device comparability and standardization. Future research should focus on direct comparisons with gold-standard methods to enhance measurement accuracy and reproducibility.

To improve the clinical and research applicability of these devices, standardized validation frameworks should be established. Strengthening the reliability, validity, and accuracy of metabolic measurements will enhance their use across diverse populations and settings, ultimately improving their utility in both clinical practice and scientific research.

## Figures and Tables

**Figure 1 sports-13-00198-f001:**
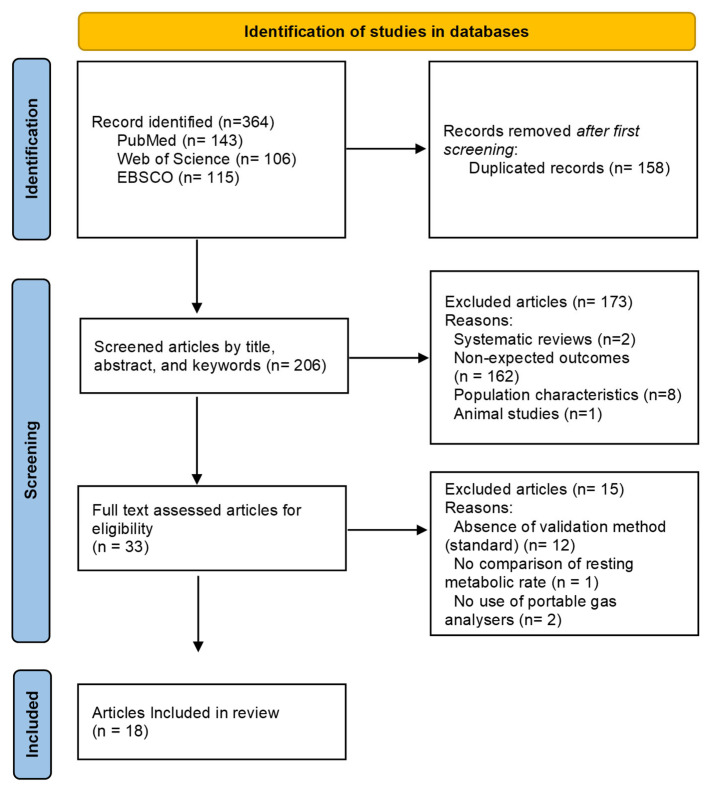
PRISMA flow diagram for the search strategy employed and results obtained.

**Table 1 sports-13-00198-t001:** Characteristics of the studies evaluating validity of resting metabolic rate using different portable gas analyzers.

Reference	Tested Device	Reference Device	Population	Variables	Protocol	Accuracy	Reliability	Conclusion
St-Onge et al., (2004) [15]	MedGem	Delta-Trac	15 (F = 6, M = 9), 36 ± 3.4 years	RMR	RMR measured for 20 min with Delta- Trac, 10 min with MedGem. Same day, random order.	No difference in RMR (6455.1 ± 417.6 vs. 6468.5 ± 337.2 kJ/d) between Delta-Trac and MedGem, respectively	Not available	MedGem is accurate for RMR comparable to Delta-Trac
Compher et al., (2005) [16]	MedGem	Delta-Trac	24 (F = 13, M = 11), 46.8 ± 15.1 years	RMR, VO_2_	RMR measurements for 20 min. Same day, random order.	Difference in RMR (1297.67 ± 202.1 vs. 1445.77 ± 285.7 kcal/day) between Delta-Trac and MedGem, respectively	No difference for reproducibility (1301.97 ± 180.9 vs. 1295.77 ± 223.4 kcal/day) Mean difference betwen measures of 6.8 kcal, with limits of agreement from 233 to 247 kcal	MedGem has adequate reproducibility, but its clinical use should be carefully considered, especially for vulnerable populations requiring precise measurements
Hlynsky et al., (2005) [17]	MedGem	Delta-Trac	F = 27, *n* = 12 subjets with anorexia, *n* = 15 control group (32 ± 8 years)	RMR, VO_2_	RMR measured using MedGem (10 min) and DeltaTrac (20 min). Same day.	Mean difference of 123.3 ± 264.5 kcal/day between Deltatrac and the MedGem. Correlations of RMR (r = 0.60, *p* = 0.04) for subjects with anorexia and (r = 0.04, *p* = 0.89) for control group	Not available	MedGem did not provide a reliable measure of RMR when compared with the Delta-Trac
Stewart et al., (2005) [18]	MedGem	Delta-Trac	50 (F = 38, M = 12) 33.8 ± 13.2 years	RMR, VO_2_	RMR measured for 10 min in a reclined position. Same day.	Mean difference for RMR mean (4.66 ± 113.39 kcal/day) (*p* = 0.773) and the correlation coefficient for RMR was r = 0.941 *p* ˂ 0.01)	Not available	MedGem measures oxygen consumption and RMR accurately where traditional metabolic carts are impractical or costly
Frankenfield & Coleman, (2013) [19]	MedGem	Delta-Trac	100 (F = 84, M = 16) 44 ± 15 years	RMR, VO_2_	RMR measured in a reclined and supine position for 10 min (Delta-Trac) and in seated position for MedGem. Same day, random order.	Difference between MedGem and Delta-Trac measurement in semi recumbent posture. Oxygen consumption (273 ± 58 vs. 247 ± 44 mL/min and No significant bias in the non-obesity group	RMR absolute difference was 61 ± 49 kcal/day. A total of 73% of the repeated measures had a 95% CI: (55–86%)	MedGem can be useful, but its accuracy varies based on obesity status, showing bias in obese individuals
Cooper et al., (2009) [20]	MedGem, Parvomedics TrueOne 2400, MedGraphics CPX Ultima, Korr ReeVue, Vmax Encore System	Delta-Trac	41 (F = 34, M = 7), 49 ± 9 years	RMR	RMR measured for 30 min on each device, excluding first and last 5 min. different day, random order.	All of the RMR CVs (Ultima 10.9%, Korr 11.9%, Vmax 8% and True One 4.8%) was significantly larger than the CV for the Delta-Trac (3%)	Not available	TrueOne and the Vmax were the most valid gas analysis systems of those tested for measuring both RMR relative to the Delta-Trac; Variability in RMR measurement consistency suggests that the choice of gas analysis system can influence results
Welch et al., (2015) [21]	ParvoMedics TrueOne 2400	Cosmed K4b2	31 (F = 13, M = 18), 27.3 ± 7.8 years	RMR, FeO_2_, VO_2_	Supine RMR measurement, 10 min rest, results averaged per minute. Different day, random order.	No significant difference in RMR (kcal/day). Difference in FEO2 (Parvo2: 19.68%, Cosmed K4b2: 16.63%),	Significant difference in measured kcal/day (*p* = 0.036) between all Cosmed RMR measurements, mean difference between Cosmed2−Cosmed1 (135.0 ± 334.7) and Cos- med2−Cosmed3 (−43.2 ± 352.7).	Due to differences in measurement technology, FEO2 was significantly different between systems, but the resultant RMR values were not significantly different
Nieman et al., (2006) [22]	FitMate	Douglas Bag (capacity not especified)	60 (F = 30, M = 30), M: 37.9 ± 13.4, F:139.8 ± 12.9 years	RMR, VO_2_	RMR measured for 10 min on both devices. Same day.	No differences between Douglas Bag and FitMate for VO_2_ (242 ± 49 mL/min vs. 240 ± 49 mL/min, *p* = 0.066) and RMR (1662 ± 340 kcal/day vs. 1668 ± 344 kcal/day, *p* = 0.579). Absolute difference 5.81 ± 80.70 kcal/day)	Not available	FitMate is a reliable and valid system for measuring VO_2_ and RMR in adults, showing high consistency with the reference method
Vandarakis et al., (2013) [23]	FitMate	Quark CPET	30 (F = 15, M = 15), 28.4 ± 7 years	RMR, VO_2_	RMR measured twice for 10 min on each device in a supine position	No differences between Quark CPET and FitMate for measured variables VO2 (r-value = 0.98, *p* = 0.0001), RMR (r-value = 0.96, *p* = 0.0001). RMR values between systems were 0.83%, mean difference of 5.95 kcal/day.	Not available	FitMate is reliable for measuring RMR in healthy adults
Purcell et al., (2020) [24]	Fitmate GS	Whole Body Calorimetry	77 (F = 49, M = 28), 32 ± 8 years	RMR, VO_2_	RMR measured using Fitmate GS (10 min) and WBC (30 min). Different days, random order.	Fitmate GS showed significantly higher VO_2_ (229 [IQR: 197–272] vs. 263 [IQR: 229–301] mL/min, *p* < 0.001). Fitmate GS underestimated RMR (1680 ± 420 vs. 1916 ± 461 kcal/day, *p* < 0.001)	RMR with Fitmate GS was of ICC 0.80 (95% CI: 0.70–0.87). Mean differences −28 kcal/day (normal or overweight) to 14 kcal/day (obesity).	Fitmate GS has discrepancies compared to whole-body calorimetry, affecting its accuracy and precision. No significant relationship between bias and body composition variables
Oshima et al., (2020) [25]	Cosmed Q-NRG	Delta-Trac, Quark RMR, V-max, ECOVX	277 (F = 128, M = 149), 67 ± 13	RMR	RMR measured for 20–30 min on all devices. Same day.	RMR differences between Cosmed Q-NRG (307.4 ± 324.5, *p* < 0.001), Quark RMR (224.4 ± 514.9, *p* = 0.038), and V-max (449.6 ± 667.4, *p* < 0.001) vs. Delta-Trac	Not available	Cosmed Q-NRG is effective and consistent for RMR measurement compared to currently used devices
Dupertuis et al., (2022) [26]	Cosmed Q-NRG	Quark RMR	85 (F = 45, M = 40), 53 ± 18 years	RMR, VO_2_	Rest time: 10–20 min, Measurement: 15 min. Same day, random order.	Higher correlation when Cosmed Q-NRG was used in canopy hood than in face mask mode (r = 0.96 and 0.86). Face mask mode overestimated RMR by 150 ± 51 kcal/day compared to canopy hood mode	Not available	Hood mode of Q-NRG is more suitable for lower-weight patients, providing precise and consistent VO_2_ measurements. Mask mode may present stability and accuracy challenges
Alcantara et al., (2022) [27]	Cosmed Q-NRG, Vyaire Vyntus CPX, Omnical Medgraphics, Ultima CardiO2	Comparison between the four gas analyzers	29, F = 11, M = 18, 24 ± 4 years	RMR, VO_2_	RMR measured for 30 min on both devices. Different days, random order.	Measurement error for RMR (Omnical = 1.5 ± 0.5%; Q- NRG = 2.5 ± 1.3%; Ultima = 10.7 ± 11.0%; Vyntu s= 13.8 ± 5.0%)	No differences (*p* = 0.058) for RMR within-subject reproducibility (inter-day CV: Q-NRG = 3.6 ± 2.5%; Omnical = 4.8 ± 3.5%; Vyntus = 5.0 ± 5.6%; Ultima = 5.7 ± 4.6%),	There is variability between devices; the Omnical device appears to be the most suitable for measuring RMR and RER
Alcantara et al., (2018) [28]	CCM Express	Ultima CardiO2 (MGU)	17 (F = 11, M = 6), 23.2 ± 2.7 years	RMR, VO_2_	RMR measured for 20 min on both devices. Different days, random order.	Mean difference between devices for RMR 65 ± 161 Kcal/day	Absolute inter-day RMR differences (158 ± 154 vs. 219 ± 185 kcal/day) or (18.3 ± 17.2 vs. 13.5 ± 15.3) between MGU and CCM.	Both devices are consistent in RMR measurement but show significant differences in their absolute values. CCM its more reliable
Wang et al., (2018) [29]	IIM-IC-100	VO2000 Medical Graphics Corp	32, F = 17, M = 15, 25 ± 6 years	RMR, VO_2_	Measurement in supine position for 15 min. for both teams. Same day random order	Mean difference between devices for RMR 81.3 kcal/d (5.83%). The CV were 5.9% and 10.3% for VO2; 5.8% and 10.5% for RMR	Significant correlations between repeated measurements for both the IIM-IC-100 (VO2: r = 0.95, VCO2: r = 0.91, REE: r = 0.95; *p* < 0.001) and VO2000 (VO2: r = 0.90, VCO2: r = 0.85, REE: r = 0.90; *p* < 0.001).	The IIM-IC-100 showed high consistency and accuracy in the measurement of RMR and RQ, comparable to the VO2000
Poulsen et al., (2019) [30]	Beacon 3	Ecovx F-CM1-04	16 M, 33 ± 9 years	RMR, VO_2_, FiO_2_	Four consecutive periods of 15 min in sitting position. at different FiO_2_ levels: 21%, 50%, 85%, and again 21%. Same day, random order.	Differences in RMR and VO_2_ between devices at differents levels of FiO_2_, especially at 85% (9%) (*p* = 0.000 for VO_2_ and *p* = 0.001 for RMR)	The CVs for EE at 21% FiO2 was Beacon 3 (4.8%) and Ecovx (4.0%)	Although both devices can be used to measure energy expenditure, differences in their results should be considered, especially in high FiO_2_ conditions, which could affect the clinical interpretation of the data obtained
Graf et al., (2013) [31]	QuarkRMR CCMexpress	Delta-Trac	24 (F = 15, M = 9), 53 ± 15 years	RMR, VO_2_	Rest time: 15 min, Measurement 10 min. Same day.	Mean RMR measured by CCMexpress canopy was (7%) higher than Delta-Trac (*p* = 0.004) The RMR limits of agreement were high (±402 kcal for CCMexpress (facemask), and ±304 kcal for CCMexpress (face tent)	Not available	Mean RMR measured by QuarkRMR is similar to Delta-Trac but the limits of agreement are high. Mean RMR measured by CCMexpress (canopy) was overestimated compared to Delta-Trac. None of the compared devices ideally replaces the Delta-Trac measurements
Zhao et al., (2014) [32]	Pocket-Sized Metabolic Analyzer	Douglas Bag (4-L)	30 (F = 15, M = 15), 27 ± 6 years	RMR, VO_2_	Collection of 4 L of exhaled oxygen while seated to calculate RMR	Significant correlation and agreement for RMR and VO_2_ between devices. Differences averaged 10%. Difference between devices for RMR 3.2%	Not available	The Pocket-Sized Metabolic Analyzer shows high accuracy for measuring RMR and VO_2_ compared to the Douglas bag

CV: coefficient of variation; F: female; FeO_2_: fraction of oxygen in exhaled air; FiO_2_: fraction of inspired oxygen; M: male; RMR: resting metabolic rate; VO_2_: oxygen consumption.

**Table 2 sports-13-00198-t002:** Methodological quality assessment tool for observational cohort and cross-sectional studies.

Reference	Q1	Q2	Q3	Q4	Q5	Q6	Q7	Q8	Q9	Q10	Q11	Q12	Q13	Q14	Score	%	Risk of Bias
St-Onge et al. [15]	Yes	Yes	ND	Yes	No	Yes	Yes	No	Yes	No	Yes	Yes	Yes	Yes	10/14	71.4	Regular
Compher et al. [16]	Yes	Yes	ND	Yes	No	Yes	Yes	No	Yes	No	Yes	Yes	Yes	Yes	10/14	71.4	Regular
Hlynsky et al. [17]	Yes	Yes	Yes	Yes	No	Yes	Yes	Yes	Yes	No	Yes	Yes	Yes	Yes	12/14	85.7	Low
Stewart et al. [18]	Yes	Yes	ND	Yes	No	Yes	Yes	Yes	Yes	No	Yes	Yes	NR	Yes	10/14	71.4	Regular
Frankenfield & Coleman, [19]	Yes	Yes	ND	Yes	Yes	Yes	Yes	No	Yes	No	Yes	Yes	Yes	Yes	11/14	78.6	Low
Cooper et al. [20]	Yes	Yes	ND	Yes	No	Yes	Yes	No	Yes	No	Yes	Yes	Yes	Yes	10/14	71.4	Regular
Welch et al. [21]	Yes	Yes	ND	Yes	No	Yes	Yes	No	Yes	No	Yes	Yes	NR	Yes	9/14	64.3	Regular
Nieman et al. [22]	Yes	Yes	ND	Yes	No	Yes	Yes	No	Yes	Yes	Yes	Yes	Yes	Yes	11/14	78.6	Low
Vandarakis et al. [23]	Yes	Yes	ND	Yes	No	Yes	Yes	No	Yes	No	Yes	Yes	Yes	Yes	10/14	71.4	Regular
Purcell et al. [24]	Yes	Yes	ND	Yes	No	Yes	Yes	No	Yes	No	Yes	Yes	NR	Yes	9/14	64.3	Regular
Oshima et al. [25]	Yes	Yes	ND	Yes	No	Yes	Yes	No	Yes	No	Yes	Yes	Yes	Yes	10/14	71.4	Regular
Dupertuis et al. [26]	Yes	Yes	ND	Yes	No	Yes	Yes	No	Yes	No	Yes	Yes	NR	Yes	9/14	64.3	Regular
Alcantara et al. [27]	Yes	Yes	ND	Yes	No	Yes	Yes	No	Yes	No	Yes	Yes	Yes	Yes	10/14	71.4	Regular
Alcantara et al. [28]	Yes	Yes	ND	Yes	No	Yes	Yes	No	Yes	No	Yes	Yes	Yes	Yes	10/14	71.4	Regular
Wang et al. [29]	Yes	Yes	ND	Yes	No	Yes	Yes	Yes	Yes	Yes	Yes	Yes	Yes	Yes	12/14	85.7	Low
Poulsen et al. [30]	Yes	Yes	ND	Yes	Yes	Yes	Yes	Yes	Yes	No	Yes	Yes	Yes	Yes	12/14	85.7	Low
Graf et al. [31]	Yes	Yes	ND	Yes	No	Yes	Yes	Yes	Yes	Yes	Yes	Yes	Yes	Yes	12/14	85.7	Low
Zhao et al. [32]	Yes	Yes	Yes	Yes	Yes	Yes	Yes	No	Yes	No	Yes	Yes	Yes	Yes	12/14	85.7	Low

ND = not determinable; NR = not reported. Q1. Was the research question or objective in this paper clearly stated?; Q2. Was the study population clearly specified and defined?; Q3. Was the participation rate of eligible persons at least 50%?; Q4. Were all the subjects selected or recruited from the same or similar populations (including the same time period)? Were inclusion and exclusion criteria for being in the study prespecified and applied uniformly to all participants?; Q5. Was a sample size justification, power description, or variance and effect estimates provided?; Q6. For the analyses in this paper, were the exposure(s) of interest measured prior to the outcome(s) being measured?; Q7. Was the timeframe sufficient so that one could reasonably expect to see an association between exposure and outcome if it existed?; Q8. For exposures that can vary in amount or level, did the study examine different levels of the exposure as related to the outcome?; Q9. Were the exposure measures (independent variables) clearly defined, valid, reliable, and implemented consistently across all study participants?; Q10. Was/were the exposure(s) assessed more than once over time?; Q11. Were the outcome measures (dependent variables) clearly defined, valid, reliable, and implemented consistently across all study participants?; Q12. Were the outcome assessors blinded to the exposure status of participants?; Q13. Was loss to follow-up after baseline 20% or less?; Q14. Were key potential confounding variables measured and adjusted statistically for their impact on the relationship between exposure(s) and outcome(s)? Background colors are presented to show a yes (green) or no (orange) answer to each item.

## Data Availability

Data are available upon request to the corresponding author (M.A.H.-L.).

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
