# Peer review of "Using Respiratory Gas Analyzers to Determine Resting Metabolic Rate in Adults: A Systematic Review of Validity Studies"

_sports, 2025, doi:10.3390/sports13070198_

Round 1
Reviewer 1 Report
Comments and Suggestions for Authors
The Validity of portable gas monitors should be summarized, so such a review is important. The review process seems good in general, but the following concerns should be addressed:
Some devices or methods such as DeltaTrac and Douglas bag methods are highly accurate in general, but their improper use can lead to systematic errors. Therefore, it is desirable that the validity and reliability of the measurements themselves be verified at each facility. Q11 in Table 1 may address this point, but has this been verified?
The phrase “reliability, validity, and accuracy” appears frequently, but how is “validity” different from “accuracy”?
L.26 and L.55
What isotope methods are used in the determination of not total energy expenditure, but RMR?
L.104
Probably "resting energy expenditure" and “basal energy expenditure” should be included as keywords. Similarly, there may be other keywords for gas analyzers that are needed.
Author Response
Reviewer 1
We truly appreciate your contributions. The changes requested can be identified in the new version of our manuscript (sports-3602879-R1) while a response to your comment is described below:
The Validity of portable gas monitors should be summarized, so such a review is important. The review process seems good in general, but the following concerns should be addressed:
Some devices or methods such as DeltaTrac and Douglas bag methods are highly accurate in general, but their improper use can lead to systematic errors. Therefore, it is desirable that the validity and reliability of the measurements themselves be verified at each facility. Q11 in Table 1 may address this point, but has this been verified?
R= According to your observation, we revised Q11 (Were the outcome measures (dependent variables) clearly defined, valid, reliable, and implemented consistently across all study participants?) carefully in all the included articles, and the outcomes variables in the studies refer to the differences between gas analyzers, so we can confirm that they used a correct methodology to validate the analyzers. We really appreciate your observation.
The phrase “reliability, validity, and accuracy” appears frequently, but how is “validity” different from “accuracy”?
R= Considering the comments of two reviewers, Reliability usually defines the reproducibility (as measured by coefficient of variance or CV), while validity and accuracy are used to denote precision similar to a standard (preferably a traceable standard or a proven reference criterion). For that reasons, we decided to stay only with the term “validity”, so it has been modified since the title. Thank you.
L.26 and L.55
What isotope methods are used in the determination of not total energy expenditure, but RMR?
R= To our knowledge, isotopic methods such as the doubly labeled water (DLW) technique are primarily used to measure total energy expenditure over a period of time in free-living individuals. These methods do not provide direct measurements of resting metabolic rate. Instead, RMR is typically assessed using indirect calorimetry, which measures oxygen consumption and carbon dioxide production under controlled resting conditions. We have clarified this point in the manuscript to avoid confusion, including the following sentence: “Several approaches are commonly used to estimate energy expenditure, including predictive equations or isotopic dilution techniques (water doubly marked with deuterium and O2), but these methods do not provide direct measurements of RMR. Instead, RMR is typically assessed using indirect calorimetry, which is considered the reference standard to measure metabolic gas exchange, as it provides direct measurements of oxygen consumption (VO₂) and carbon dioxide production (VCO₂) under controlled resting conditions”. We sincerely appreciate your observation.
L.104
Probably "resting energy expenditure" and “basal energy expenditure” should be included as keywords. Similarly, there may be other keywords for gas analyzers that are needed..
R= We have performed a new search with the proposed keyword, resulting in 60 new articles. After screening, we found two articles that met the eligibility criteria and were added to the results. The document has been substantially improved by adding the proposed keywords, thank you.
According to your observations, the article has been improved substantially. We really appreciate your contributions.
Reviewer 2 Report
Comments and Suggestions for Authors
The aim of the article is very interesting and appealing.
Thera are 2 "table 1", the first one is very difficult to "read".
Most of the included studies involve healthy adults; The subjects with metabolic diseases are less represented, in this form the results are more generalized and less rappresentative of clinical populations.
There are few differences in measurement protocols (duration, subject location, environmental conditions). It is a pitty because the device models analyzed may have contributed to the variability in results between studies.
There is not meta analysis, why?
Comments on the Quality of English Languagegood
Author Response
Reviewer 2
We truly appreciate your contributions. The changes requested can be identified in the new version of our manuscript (sports-3602879-R1) while a point-by-point response to your requests is described below:
The aim of the article is very interesting and appealing.
R= Thank you.
Thera are 2 "table 1", the first one is very difficult to "read".
R= Thank you for pointing out this error, the results table corresponds to Table 1 and the methodological quality/risk of bias to Table 2.
We appreciate your comment and understand your concern about the clarity of Table 1. We have reorganized all this table to condense the information extracted from the results of each study analyzed with clarity. We have divided a column for the tested device, a column for the reference device, and separated the results in a column of accuracy and a column for reliability.
Most of the included studies involve healthy adults; the subjects with metabolic diseases are less represented, in this form the results are more generalized and less representative of clinical populations.
R= Validation studies are generally conducted in populations of high-performance athletes or healthy individuals. Indeed, there is a deficiency in representative studies of populations with metabolic diseases. However, some gas analyzers included in the present systematic review present results in populations with anorexia and with excess body mass, but according to your observation, we have added the following sentence in the discussion section: “There were found only two studies with populations with risk factors of metabolic diseases, Purcell et al. reported differences at evaluate RMR in normal/overweight subjects compared to obesity, and Hlynsky et al. evaluated women with anorexia, reporting statistical differences of gas analyzers at comparing this population with a control group. Is important to mention that there has been reported that metabolic diseases can affect RMR by either increasing or decreasing it, making difficult an evaluation, and there exists a need to implement studies focused in describe the best equipment or method to avoid bias at evaluating energetic metabolism in subjects with metabolic diseases“.
There are few differences in measurement protocols (duration, subject location, environmental conditions). It is a pitty because the device models analyzed may have contributed to the variability in results between studies.
R= There is no standard for conducting RMR measurements; each manufacturer has its own protocols, including calibration, rest time before the test, subject position, and measurement time. For this reasons it is not usual that the authors report this variables, reporting only calibration previous each evaluation but not specifying the characteristics of calibrations. This has been mentioned in the Limitations section of the study. Thank you.
There is not meta analysis, why?
R= Dear Reviewer, due to insufficient data available for Pocket-sized Metabolic Analyzer, IIM-IC-100 VO2000, and Ecovx Beacon, we were unable to pool all the data together into a robust meta-analysis. Besides, the included studies made use of distinct metrics for representing the validity of the assessed devices (i.e., mean differences, LoA, ICC, etc), hindering the comparison among studies results. We have addressed this as a study limitation, recommending that future studies follow the statistical guidelines from Hopkins et al., (2009) for standardizing the validity assessment of portable gas analyzers.
According to your observations, the article has been improved substantially. We really appreciate your contributions.
Reviewer 3 Report
Comments and Suggestions for Authors
Please see attached

Author Response
Reviewer 3
We truly appreciate your contributions. The changes requested can be identified in the new version of our manuscript (sports-3602879-R1) while a point-by-point response to your requests is described below:
The review by Onivas-Leόn et al was intended to address the limited information about portable metabolic measurement systems for assessing resting metabolic rate (RMR). The authors correctly pointed out the challenges of accurate and reliable measurements of RMR and followed the PRISMA guidelines to conduct the “systems review” of this topic. Despite of their admirable efforts, only 16 studies with original data were included. There were several critical concerns that need to be addressed before this review is considered valuable or useful:
- The title (and throughout the paper), the three terms “Reliability, Validity, and Accuracy” were used without clear definition what they represent. Reliability usually defines the reproducibility (as measured by coefficient of variance or CV), while validity and accuracy usually define how much the measured values differ from the truth (preferably a tracible standard or a proven reference criterion). There was little reliability data presented, so this term should be eliminated. However, a good systematic review should include any test retest reproducibility data for each study. If there are not available, state as such.
R= Considering the comments of two reviewers, for the reasons that you mention, we decided to stay only with the term “validity”, so it has been modified since the title. Thank you.
- There should be a section describing the technology of the different portable analyzer systems in more details, including the manufacturers’ information, the type of measurements (O2, CO2, flow, temp, humidity, any correction for STPD), if there are still on the market, how are they calibrated, what are the manufacturer’s claimed accuracy and testing ranges (temp, humidity), what gas sampling method were used (mask, hood, or mouthpiece), how long are the recommended measurement periods, and at what posture (sitting, reclining, supine), the estimated costs of the system (and each testing kit if used), how many systems were tested, how are the criterion method (DTC or Douglas Bag) calibrated, etc. This would be critical for readers.
R= There is no standard for conducting RMR measurements and each manufacturer has its own protocols, including calibration, rest time before the test, subject position, and measurement time. Unfortunately, this usually is not reported in the studies. All this have been mentioned in the Introduction and Limitations section, including the following sentences:
“RMR is typically assessed using indirect calorimetry, which is considered the reference standard to measure metabolic gas exchange, as it provides direct measurements of oxygen consumption (VO₂) and carbon dioxide production (VCO₂) under controlled resting conditions. Traditionally, metabolic carts have been used for this purpose using different sensor technologies, calibration procedures, or data-processing algorithms. However, variations in these methodologies make it difficult to standardize indirect calorimetry protocols across different clinical and research settings. Furthermore, metabolic carts require controlled laboratory conditions, limiting their feasibility for widespread use in non-laboratory settings”.
“Furthermore, because the included studies used distinct metrics to represent the validity of the evaluated devices (i.e., mean differences, LoA, ICC, etc), this made it difficult to compare the results of the studies. Moreover, it was not possible to pool all the data in a robust meta-analysis due to the lack of available data for the Pocket-sized Metabolic Analyzer, IIM-IC-100 VO2000, and Ecovx Beacon. Finally, the analyzed findings suggest that the characteristics of the studied populations, the technology of each device, the type of measurement, the sampling method, the test range, and the calibration method can influence the accuracy of device measurements, highlighting the importance of conducting studies following the statistical guidelines of Hopkins et al. [40] in diverse populations to standardize the assessment of the validity of portable gas analyzers”.
- Table 2 lists the summary findings from the 16 studies, which appears to be chronologically ordered. It would be more helpful to reorganize this table by following:
- Separate by portable sensors (MedGem, FitMate, multiple systems, etc)
- Column 1: List authors (last name of the first author) then years (e.g., rather than [14] (USA), please use St-Onge et al (2004) [14])
- Column 2: tested device (e.g. MedGem, and version details if available)
- Column 3: reference device (e.g. Delta-Trac, and version details if available, size of the Douglas bag volume, how many samples at what time intervals)
- Column 4: population (as listed currently)
- Column 5: variables (as listed)
- Column 6: protocol (as listed, please add if tested on the same day or different days, and if the tests were randomized)
- Column 7: accuracy (present the accuracy in absolute and/or percent errors rather than kJ or kcal/day).
- Column 8: reliability (if available)
- Column 9: conclusion (as listed currently)
- Please do not overuse abbreviations (e.g., VE, FR, WBC), double check the units (e.g., the second study [15], mean age in years?) and consistency ([26], age mean >60 years?).
R= We have developed a new Table 1 following your observations, and we are sure that with this modifications the quality of it has been improved substantially, thank you.
- Table 2 (not Table 1 as labeled) starting after line 145 is perhaps less important for readers as for the authors (to conduct reviews).
R= We have corrected the Table number, and we agree that the importance of Table 2 is not the same than Table 1, but according to the PRIMSA Checklist and the reviewers suggestions, it has been presented in the document with the improvements suggested by the reviewers. Thank you.
- Please provide a reference for Douglas Bag (not “Bad” as it was spelled in line 33) as the “gold-standard” RMR measurement. Also, Douglas Bag is a general term, with the gas analyzers used may be different.
R= According to your observation we have added information in the Introduction section regarding Douglas bag, including the following sentence: “One of the earliest methods developed to determine the whole body metabolic rate was the Douglas bag, that consists on evaluate expired gases collected in an exhaled breath over a specific period in a twill bag lined with vulcanized rubber and is considered a gold standard at measuring respiratory gas exchange, but has some limitations such as a limited sample due the test time in each subject, air leaks or condensation in the bags, and the lack of an standard reference for validation”.
- It would be helpful to discuss what clinically meaningful accuracy and/or reliability thresholds are for these portable systems, what systems are now on the market but have not been validated vs. have been validated but now are not on the market, how future systems should improve on, and how validation study should be performed (what are the current knowledge gaps).
R= We have added some paragraphs to the Discussion section related to your observation, including the following: “Currently, there are various devices on the market claiming to be of high quality for assessing metabolic gas exchange, however, there is little evidence on their validity and reproducibility, and the existing evidence is focused on specific populations such as high-performance athletes and hospitalized patients. An example of this is the BIOPAC GASSYS3 which presents significant differences in its measurements for RMR in specific populations, requiring to develop equations to correct the data, which calls into question its validity as an assessment device”.
“Devices such as Delta-Trac and VO2000 have proven their validity compared to standards; however, their manufacture is discontinued, making them impossible to acquire. Likewise, there are devices that have implemented technological improvements compared to previous versions, such as Cortex MetaMax 3B or Cosmed K5. Although these updates are nearly a decade old, these devices remain reliable, but low demand and high costs could hamper the development of new devices of this type. In addition, several portable systems have been developed designed to provide basic V̇O2 measurement and RMR estimation. However, these devices have limitations, as they only provide O2 analysis and require assumptions about RMR. These portable devices are unlikely to be accepted in high-quality research where direct measures of these variables are required”.
According to your observations, the article has been improved substantially. We really appreciate your contributions.
Round 2
Reviewer 1 Report
Comments and Suggestions for Authors
The responses by the authors and the corrections are generally appropriate, but just one point still needs to be revised.
The proposed sentences on isotope dilution techniques are very confusing. The DLW method does not provide RMR, while prediction equations for RMR provide RMR values. These are sentences on RMR, not total energy expenditure. Therefore, the descriptions on isotope dilution techniques should be removed.
Author Response
May 28th, 2025
Reviewer 1
Thank you for the time devoted to our article entitled: “Validity of Respiratory Gas Analyzers to Determine Resting Metabolic Rate in Adults: A Systematic Review”, authored by César Ulises Olivas-León, Francisco Javier Olivas-Aguirre, Isaac Armando Chávez-Guevara, Horacio Eusebio Almanza-Reyes, Leslie Patrón-Romero, Genaro Rodríguez-Uribe, Francisco José Amaro-Gahete, and Marco Antonio Hernández-Lepe (corresponding author). We truly appreciate your contributions and the observation requested can be identified in the new version of our manuscript (sports-3602879-R2) while a response to your comment is described below:
The responses by the authors and the corrections are generally appropriate, but just one point still needs to be revised.
The proposed sentences on isotope dilution techniques are very confusing. The DLW method does not provide RMR, while prediction equations for RMR provide RMR values. These are sentences on RMR, not total energy expenditure. Therefore, the descriptions on isotope dilution techniques should be removed.
R= We apologize for the confusion caused by this sentence, and at your recommendation, the description of isotope dilution has been removed. The resulting sentence is the following: “Several approaches are commonly used to estimate RMR, including predictive equations, but this method does not provide direct measurements of RMR. Instead, indirect calorimetry is often used to determine RMR”.
According to your observations during the review process, the article has been improved substantially. We really appreciate your time and support, sincerely
Dr. Marco Antonio Hernández-Lepe, Corresponding author
Reviewer 3 Report
Comments and Suggestions for Authors
Thank for responding to my comments.
Author Response
Comment: Thank for responding to my comments.
R= We really appreciate your contributions.